# The Challenges of Promoting Social Inclusion through Sport: The Experience of a Sport-Based Initiative in Italy

**Chiara D'Angelo** *, **Chiara Corvino** and **Caterina Gozzoli** *

Department of Psychology, Catholic University of the Sacred Heart, 20123 Milan, Italy; chiara.corvino1@unicatt.it
* Correspondence: chiara.dangelo@unicatt.it (C.D.); caterina.gozzoli@unicatt.it (C.G.)

**Abstract:** Social inclusion is broadly recognized as a priority to accomplish at an international level. While the influence of sport toward this social mission has been largely debated, literature lacks contributions capturing the challenges of sport when promoting social inclusion. Based in case study methodology, the investigation explores the impact of a multi-stakeholder sport initiative developing social inclusion for socially vulnerable youth and the related challenges of the intervention through in-depth interviews with diverse program stakeholders. The main findings indicated the emergence of four challenges: limited transferability of program outcomes for youth in living conditions of severe vulnerability; drop-out of youth in living conditions of severe vulnerability; limited sustainability of program social workers; lack of sports club management skills. The work highlights some limits of sport-based programs for social inclusion and discusses some implications for practice to maximize the societal impact of such interventions.

**Keywords:** sport-for-development; sport-based program; social inclusion; socially vulnerable youth; Italy



## 1. Sport for Promoting Social Inclusion of Vulnerable Youth: An Overview

Currently, social exclusion is fully recognized at an international level as a social concern to be addressed. The 16th sustainable development goal of United Nations quotes the promotion of inclusive societies as a fundamental priority to accomplish within 2030. In recent years, the contribution of sport toward the achievement of this social goal has been largely argued, with several academics underlining that sport can be used as a vehicle to address certain aspects of social inclusion [1–6]. If, on the one hand, there is substantial agreement on the positive impact of sport at a societal level, academics argue that more research is needed in order to understand the conditions in which sport may act on social inclusion [5–11]. The investigation of such conditions is still in its infancy [12–15] and would provide a strategic and agency-focused approach for further planning of sport-based interventions [16–23].

When talking about socially vulnerable youth, literature generally refers to individuals between the age of 11 and 24 years of age who are subjected daily to multi-faceted stressors (e.g., social, emotional, and economic) that create conditions for social maladjustment [24,25]. These conditions include: (i) living in areas of low socioeconomic status and poor housing quality; (ii) receiving residential care or nonresidential counseling [26]; (iii) poor family management; and (iv) peers engaged in deviant behaviors [27,28].

In line with what has been previously exposed, literature on sport and socially vulnerable youth largely confirmed the fact that sport may develop positive social outcomes that can be associated with social inclusion, such as life skills, positive psychological capital, active citizenship, pro-social behaviors, and employment. Furthermore, in line with the main indications provided by international academics, the majority of studies have focused on understanding the conditions of sport that promote the achievement of social inclusion outcomes [17,18,22,23,26,29,30].

Recently, for instance, Hermes and colleagues [26] highlighted whether and how sport-based interventions can develop life skills for socially vulnerable youth.

The work highlights the prominent role of the sports environments created by the staff in the development of certain skills, such as self-motivation, goal setting, self-direction, critical thinking, self-concept, self-efficacy, resistance skills, concentration/attention, self-control, taking responsibility for one's own actions, and discipline. From their work, it emerges that each program is unique and, thus, may develop different outcomes according to the specific sport environment conditions promoted.

Similar considerations have been suggested by Super and colleagues [31], who noted in their research that the positive outcomes promoted by sport may have been influenced by exposure to a positive motivational climate in sport. They report that it is possible that the positive climate experienced by young people affects the development of social outcomes rather than simply participation in sport.

A recent review on Positive Youth Development (PYD) through sport by Holt and colleagues [32] also highlighted the relevance of a positive climate within the micro sport system as a condition to promote developmental outcomes for youth through sport. Holt et al. refer to this by using the term "PYD climate." Holt et al. explain in [32] (pp. 37–38):

This PYD climate is the social environment—based on relationships with and between adults, peers, and parents—that enables youth to gain experiences that will contribute to PYD outcomes.

According to Coackley [33], in order to create such a positive setting, adults should provide young people with physical safety, personal empowerment, economic support, and personal and political empowerment. In doing so, sport workers should be familiar with adequate pedagogical strategies, as well as forming positive and trusting bonds with and between their athletes.

All such contributions suggest that the creation of a positive sport environment is a fundamental condition of work if one wants to promote positive social outcomes through sport for socially vulnerable youth.

Other authors suggested that, in order to promote positive social outcomes through sport, it is important to work through a multi-agency approach [4,23,34].

Various studies have investigated the relevance of sport agencies and social institutions as delivery channels of local sport-related development [32,35]. Such collective action is particularly effective when implementing programs for socially vulnerable populations. The tackling of social fragility actually requires the sharing of technical, didactical, and educational skills emanating from different disciplinary fields and professional expertise [36,37].

Spaaij [34] noted that, when sport-based programs collaborate with community agencies, it is possible to provide young people with increased social resources, support, and opportunities in terms of employability. He also argues that youth development is meaningfully associated with a program's ability to develop connections with multiple institutional agents, thereby enabling young people to expand their social horizons.

As also noted by Jones and colleagues [38], youth sport programs should provide collaboration with community organizations by sharing resources, knowledge, and expertise through an integrated curriculum.

The capacity to provide emotional and psychological support, as well as the ability to analyze, interpret, and effectively act to address youth needs, falls within the competencies and professional range of social workers. With sport and social agencies working together, vulnerable youth will receive the necessary comprehensive care, support, and services [23].

While literature is starting to understand the conditions that may lead to the development of social inclusion through sport and related activities for socially vulnerable youth [26]—such as the creation of a positive sport environment and the adoption of a multi-agency approach—the challenges and obstacles that arise when promoting social inclusion through sport-based programs remain unexplored.

In addition to this, practitioners still seem to be anchored to an idealized image of sport as a magical tool for tackling serious social marginalization. Although, on the one

hand, researchers are aware that sport can act only on certain aspects of inclusion, on the other hand [4], there is a lack of scientific reflections involving those who work on the field for dwelling on the challenges of sport in promoting inclusion. This knowledge gap is particularly felt within the Italian context, where there are still few studies concerning sport for development and peace [22]. Faced with many sports projects dedicated to promoting social inclusion, in fact, there is still limited research dedicated to the topic [23].

In order to overcome this gap and provide a nuanced image of the potential of sport, the current research seeks to explore the challenges that may occur when using sport as a tool to promote social inclusion. In particular, the research focuses on the experience of three Italian sports clubs that were involved in a sport and social inclusion project for socially vulnerable children.

## 2. Materials and Methods

The work is based in case study methodology [39] and focuses on three grassroots sports clubs located in the province of Milan (the North of Italy) that took part in a sport-based program aiming to promote social inclusion among socially vulnerable young people.

A case study is an intensive investigation of a single case where the purpose of that study is—at least in part—to shed light on a larger class of cases (a population) [40]. This case aims to provide insights about the larger class of sport-based programs promoting social inclusion for socially vulnerable youth.

Despite the current study involving three sports clubs, the study will be treated as a single case because the three realities shared common characteristics (e.g., same professional involved, common project manager, common donor, common criteria selection for participants) that led us to consider them as a comprehensive case study. The manuscript analyzes the experience of these sports clubs through in-depth interviews with the aim of exploring the challenges that may occur in a sport-based program when using sport as a tool to promote social inclusion for socially vulnerable youth. The case, thus, is atheoretical [41] since it aims to explore these challenges without previous theoretical hypotheses grounding the study. In doing so, the current work seeks to answer the following research question: what are the challenges of developing social inclusion for socially vulnerable youth through sport-based programs?

### 2.1. The Case under Analysis

The sport-based program analyzed in this case study provided two weekly soccer training sessions (90 min each) and a total of 10 h of workshop time for transversal skills, which development participants were able to select from a list of options (pastry, cooking, mosaic, and music) based on their personal interests.

Each club selected an ad hoc soccer team for the participants to join by informal agreements with middle schools. These teams had access to the specially qualified staff employed. Each team had access to a sport coach (offering regular training), an educator (co-managing the training and providing regular feedback to the youth), and a psychologist (in a supervisory and advisory capacity).

During the three years of program implementation, the youngsters attended an average of 200 sessions of soccer training and 20 sessions of transversal skills development.

In these clubs (whose anonymity has been preserved) a total of 49 young participants took part in the soccer activities. All participants were aged between 11 and 15 years (with an average of 13 years) and most were several-generation Italian (75%); a minority (25%) were second-generation Italian. Inclusion criteria for program participants were: poor school attendance, poor academic performance, lack of family interaction with the school, a tendency to break school rules, and deviant behavior or a tendency toward relational isolation (as reported by the schools). Thus, young people recruited for the program experienced various psychological and/or social problems (from high to moderate vulnerability) that prevented them from taking full advantage of their relationships with

significant others and their peers in the school environment—a condition that contributed to their level of vulnerability as reported in Table 1.

**Table 1.** Characteristics of participants.

| | No. | Drop-Out during the Three Years | No. Youth Living Conditions of High Vulnerability | No. Living Conditions of Moderate Vulnerability |
|---|---|---|---|---|
| Sport Club A | 22 | 0 | 4 youth were living with dysfunctional parenting and/or absence of caregivers, deviant behavior, and a tendency to break school rules with a high chance of dropping out of school. | 18 youth were exposed to moderate vulnerability conditions, such as relational challenges at schools (e.g., challenges in relating with peers; conflicts with teachers) and economical poverty |
| Sport Club B | 16 | 0 | 3 youth showed a tendency to break school rules with a high chance of dropping out of school. | 13 were exposed to low vulnerability conditions such as relational challenges at schools |
| Sport Club C | 11 | 7 | 2 youth were living with dysfunctional parenting and/or absence of caregivers, 4 youth showed deviant behavior at school1 youth was victim of bullying at school | 4 youth were exposed to moderate vulnerability conditions such as social isolation at school |
| Total | 49 | 7 | 14 | 35 |

The educators and psychologists involved in each soccer team assisted in the recruitment of participants. They also helped with stakeholder liaison, a process aimed at connecting the club with schools and social/health service providers (thus, forging strong and multiple-stakeholder relationships). Schools also engaged in networking through the program. Consistent with the principle of interconnectivity, staff members (educators and psychologists) implemented activities for networking with civic and government agencies. This included meetings with local municipalities, youth associations, and educational centers (e.g., day centers). Staff held monthly meetings with target schools during the final two years of program implementation. In Club C, five participants encountering high risks (e.g., parents in prison, neuropsychological disorder[s], etc.) were referred for specialized assistance; this was monitored by the program staff.

Consistent with the principles of positive youth development, the program aimed to develop social inclusion through diverse strategies:

- creation of a positive climate by the sport coach leading to the development of individuals' sense of acceptance and perception of inclusion;
- positive communication and positive feedbacks by the sport coach strengthening participants' self-efficacy in sports;
- individual meetings between the psychologists and participants for overcoming individual challenges during sport training;
- collaboration with youth school teachers.

*2.2. Procedure*

Participants were involved in the research process at the end of the three-year project with the intention of reflecting back on the strengths and weaknesses of the activities.

Each participant was personally contacted by the researcher in order to show the purposes and procedure of the investigation. Participants who agreed to be interviewed were asked to sign a written consent form.

The authors declare that the procedure met the international norms and ethical principles established by the European Union 2016/679 Regulation (UE, 2016), the Declaration of

Helsinki (World Medical Association, 1964) and related revisions, with written informed consent obtained from each participant.

### 2.3. Data Analysis and Tools

As previously reported, the aim of the current investigation was to explore the challenges that may occur in a sport-based program when using sport as a tool to promote social inclusion for socially vulnerable youth. Within the interviews, this specific research aim was investigated by asking diverse types of questions, including a number related to the specific inclusion impact of the program. This choice was dictated by the fact that it would have been difficult to address the challenges and difficulties of the program during the interview without an in-depth understanding of the social inclusion impact of the activities. Due to this, the interviews guided the following areas of investigation:

(i)   Impact of the program in terms of social inclusion (e.g., how did the program affect the social inclusion of the youth involved? What features of the program contribute to address social inclusion through sport?)

(ii)  Challenges when promoting social inclusion through sport for socially vulnerable youth. (What challenges did you face during the three years of program implementation? What features of the program were most challenging?)

The interviews lasted about 45 min to an hour; they were conducted at the sport clubs (with the exception of the teachers, who were interviewed at school). All material was analyzed by an inductive content analysis bottom-up approach, where themes emerged directly from the data [42]. The trustworthiness and validity of the data analysis was guaranteed by a process of data triangulation [43]. We triangulated themes that emerged through different sources of interviews (sport workers, social workers, teacher,; the project manager, and donors). Communalities, divergences, and connections across themes were pointed out and discussed by two independent researchers (CD–CC), who read the material and compared main themes emerging from each source of data. Divergences were discussed until agreement was achieved.

### 2.4. Sampling

Interviewees included soccer club coaches (4) and administrators (4), educators (3) and psychologists (3), donor foundation managers (2), the project manager (1), and teachers (4), for a total of 21 interviews collected as reported in Table 2. The mean age of participants was 40. Interviewees involved in the research were selected according to the logic of purposive sampling.

**Table 2.** Sample of the research.

| Participants | Sport Club A | Sport Club B | Sport Club C | Total |
|:---:|:---:|:---:|:---:|:---:|
| Sport coaches | 1 | 1 | 2 | 4 |
| Sport administrators | 2 | 1 | 1 | 4 |
| Educators | 1 | 1 | 1 | 3 |
| Psychologists | 1 | 1 | 1 | 3 |
| Teachers | 0 | 3 | 1 | 4 |
| Donors [1] | | 2 | | 2 |
| Project Manager [1] | | 1 | | 1 |
| | | Total: 21 | | |

[1] Donors and Project Manager were the same for the three clubs.

## 3. Findings

The results are summed up in Table 3 and highlight the impact of the project in terms of social inclusion and the related challenges reported by the respondents.

**Table 3.** Results.

| Program Impact | Challenges |
|---|---|
| Improved youth self-efficacy<br>Development of youth social capital<br>Development of social capital at the community level serving the social inclusion of youth | Limited transferability of program outcomes for youth in living conditions of severe vulnerability<br>Drop-out of youth in living conditions of severe vulnerability<br>Limited sustainability of program social workers<br>Lack of sports club management skills |

*3.1. Program Impact*

3.1.1. Improved Youth Self-Efficacy

Interviews highlighted that the program provided insecure young people with a positive environment in which to gain more confidence in their abilities, especially regarding their capacity to relate with peers. Improved self-efficacy was also associated with a general improvement at a physical level. For instance, program workers in sport club C reported the case of a boy who had great difficulties in interacting with his peers and properly using his corporeal skills during soccer training. At the end of the project, both social workers highlighted a change at an individual level related to improved self-efficacy and confidence during training and teachers pointed out improved social interactions at school, as described by these interviewees:

> "He was clumsy, sometimes almost catatonic, in front of the ball: he did not know what to do and the project helped him in terms of self-efficacy." Educator (sport club C)

> "When he arrived here, he found it hard to look you in the eye, he hardly talked. If you looked at him, he became intimidated and stopped. He developed effective skills from a motor point of view, he had the chance to meet youths like him. He found a number of conditions that made him say "wow, I'm not alone in the world!"" Psychologist (sport club C)

> "He had great difficulties in relating to the rest of the class, and therefore, surely it can be seen that he improved on both sides [the teacher is referring to school and sport settings]. He is more outgoing, more involved, this is a boy who could barely say his name and surname. Now he is more extroverted; he participates more in the didactic activities, and this is also underlined by the educator and psychologist of the program. Thus, we can say that the program was a full success for him." Teacher (sport club C)

3.1.2. Development of Youth Social Capital

The program was also a resource for establishing positive relationships for the young people who were socially isolated at the baseline. For instance, in sport club B, the middle school recruited a migrant youth who had recently arrived in Italy. The positive environment created by program workers supported him to learn Italian and engage in relationships with Italians peers, as described in these interviews:

> "He learned Italian with us and found new friends; he established a trusting relationship with A; they even meet outside the sport context. Thus, his case is an example of fighting social isolation." Educator (sport club B)

> "His participation in the project aimed to bring him closer to new relationships, and to the language also. And here, too, we had positive results." Teacher (sport club B)

3.1.3. Development of Social Capital at the Community Level Serving the Social Inclusion of Youth

One of the innovative features of the program was the implementation of meetings between the program's staff with the middle-school teachers who recruited the participants.

The purpose of these meetings was to discuss the educational progress of participants in the sports program, as well as their educational path at school. The results highlight that such collaboration resulted in improved educational work with the young people.

> "I liked the way we worked together [the teacher is talking about the meeting with social workers] because we really worked together, I mean, we had significant exchanges of observation and understanding about the young people that could lead teachers to reflect on themselves and to observe something new in their students." Teacher (sport club B)

> "If there was something wrong with the participants and they were not aware of the reasons for it [the teacher is referring to a program educator and psychologist], we informed them by reporting why, in that situation, participants were a little nervous. We told them everything." Teacher (sport club C)

Meetings between program social workers and teachers each month provided teachers with the opportunity to know the participants from another perspective (by learning how they were behaving within the sport clubs).

In contrast, program educators and psychologists could better understand youth behaviors and attitudes by comparing what teachers reported during meetings. Thus, the sharing of information was useful for attaining a deeper and more integrated understanding of the young people's performance of the educational work given by teachers and the program of social workers.

In some cases, this exchange allowed teachers to change their stereotyped view of particular vulnerable children, as described by this interviewee:

> "I knew from program educator that this girl did a very good job in creating the plot of the video. She mainly wrote down the plot, and she even took the job home to finish it, to improve it. It has been a real success." Teacher (sport club C)

Even if sport workers did not directly participate in meetings with the schools, in some cases, the information shared in the meetings was disseminated among the sport club professionals. For instance, a sport administrator claimed that connecting with schools was quite effective as it enhanced his awareness and consciousness of the young people's needs outside the sport environment:

> "The constant dialogue with the schools this year was very helpful because they gave us a different perspective. We used to see the guys only from an athletic point of view, and teachers showed us a different vision: kids don't do just sport, they experience fatigue due to studying, and they have difficulties sitting for five hours at school, which made us realize that we did not consider a number of facts before." Sport administrator (sport club C)

Furthermore, such meetings guaranteed mutual help among the network of adults. Teachers started requesting the support of program staff if youngsters were in trouble. For instance, this teacher explained that she asked for the program psychologist's help in coping with a situation she could not manage on her own:

> "A program participant had an argument with her speech therapist and decided to stop her therapy with her. It was a very delicate situation, so I talked with the girl, and at the end of our chat, I told her I was going to call the psychologist of the sport program. I thought he was more skilled than me in coping with this situation. I told her he could help her better than me." Teacher (sport club B)

In sport club C, teachers even created linkages between the program psychologist and educators and public service providers who assisted youth who needed special help (for instance, psychological support):

> "I will give you an example, two program participants were suspended from school activities for three days because they assaulted a guy. Then their teacher called me and said, "Can I give your phone number to their psychologist?" I

answered, "yes, of course, give her my number." The psychologist calls me and explains the situation to me and asks me for further information about the girl, and then she says to me: "Listen, tomorrow, we have a meeting with the social service providers taking care of the girl; do you want to come to the meeting?" I couldn't go to the meeting, but I told her to keep in touch in order to better understand how to manage the situation. How was I helpful in this case? I gave her my point of view on the relationship of the mother with the daughter, the relationship of R. with her mother and the relationship with the school." Psychologist (sport club C)

One teacher also claimed that the connection between the sporting context and schools was an important added value for the common good. In fact, the program started to be viewed as a valuable resource for the educational community, as reported by these participants:

"Because of the project, we have now strengthened the ties, maybe we have created a triad with the project because we know that there is also this project that can intervene in support of this girl or that boy in need." Teacher (sport club B)

"Because of the project, we are more present as an educational body that can provide support for the school. While this was not the case before, the kids went to school and then they practiced sport, but we had never thought about talking to each other before." Program Educator (sport club C)

Such findings suggest that the collaboration between program workers—who were observing the youths inside the sport environment—and teachers—who were aware of youth needs and challenges at school—permitted the formation of a broader and less stigmatized image of the young people, which serves the understanding of their progress at an educational level. Furthermore, such a collaboration provided the young people with increased emotional and social resources useful in overcoming their vulnerabilities.

*3.2. Challenges*

3.2.1. Limited Transferability of Program Outcomes for Youth in Living Conditions of Severe Vulnerability

Although data shows the positive impact of the program in terms of self- efficacy and social capital development, teachers and educators reported that the project did not have any kind of impact at the school level for most vulnerable youth.

"I would say [the teacher is talking about the main results of the project] acceptance and understanding of the social norms. In some cases, they have not been achieved—for some participants we cannot say that they [teacher is talking about program outcomes] have been completely achieved; especially with regard to respecting rules." Teacher (sport club B)

For instance, in sport club C during the third year of implementation, the young people were involved in a video-making workshop. They were asked to structure and act in a plot for a video. The plot of the video was centered on a boy who was a victim of bullying who was gradually able to develop skills and ultimately emancipate himself through sport.

One girl, who was a bully at school, had the chance to play a positive leadership role in the group during the making of the video as she devised the plot. Although she became more aware of her behaviors, in the end she did not change her actions at school, because her family situation was particularly risky from a psycho-social point of view, as explained by the program psychologist.

"She is very seductive, she is very manipulative, she is very borderline, she has a series of problems [the psychologist is referring to the fact that the girl has an absent father] that also make her think a bit. The project is giving her a regulatory container that she needs that is serving her a lot, because she finds her

own dimension. It is very useful to her in individual terms to have a space and in regulatory terms to learn to confine her exuberance, in terms of role models because she doesn't have any, and therefore, she is growing a lot from this point of view. At the moment, this does not have a huge impact on the school because the problem is much more complex, as also reported by her teacher." Psychologist (sport club C)

It emerges that, when the social vulnerability of youth is very high, the benefits of sport are hardly transferred at the school level.

### 3.2.2. The Challenge of Youth Drop-Out in Living Conditions of Severe Vulnerability

Participants reported challenges in terms of youth recruitment. During the first year of the program, school leaders and teachers sent a relevant percentage of highly vulnerable youth to program activities in sport club C that impacted the work of the coaches and social workers, who often had to deal with episodes of violence and fighting among the young people. Interviewees reported several episodes where the young people did not respect the rules during sport training.

"They [the educator is referring to program participants] are quite agitated. During the first training, it was tiring. They don't respect the rules, they do not respect the role, even with me they have exaggerated several times. Last time, they started kicking balls at the sport coach who was preparing the football goal and insulted him." Educator (sport club C)

Thus, the youngsters' behaviors impacted the delivery of the activities and the rate of participant dropout. Nevertheless, the soccer team was comprised primarily of highly vulnerable adolescents, and as such, they were at risk of reiterating mechanisms of exclusion and stigmatization. These features were highly challenging; they led to a second round of recruitment in order to recruit more widely during the second year of the program. As explained by the following donor, it was relevant to better define the target population and avoid recruiting mostly highly vulnerable youngsters:

"During the second, year we focused better on our own target. The recruitment of the boys was done in a slightly more structured way, that is, we talked with schools about dropping out in a broad sense, that is, we asked them not to only send us people who dropped out of school. They started sending us who was at risk, students who they saw as possibly at risk." Donor foundation manager

### 3.2.3. Limited Sustainability of Program Social Workers

In sport clubs B and C, educators were crucial actors that facilitated the clubs' connections with the middle-school level. Interviewees reported that they were responsible for connecting with the schools and understanding what was happening there, while coaches were asked to work on the ground with youth.

Data showed that the role assumed by social workers obstructed sport clubs' leadership in networking with schools and being active interlocutors with teachers. Indeed, teachers reported being connected to program educators and psychologists rather than the sport clubs.

"Absolutely not, I haven't seen them [interviewer is talking about the networking of the program and is referring to the relationship with members of the sport club]. I haven't relationships with the soccer clubs, no, we are not in a network with them." Teacher sport club C

"The main interlocutors for the program were the program educator and psychologist." Teacher (sport club B)

In terms of sustainability, this aspect was critical. Once the funding of the foundation ends, the network built by social workers is at risk for collapse:

"Building this kind of network [interviewer is referring to the connection with the middle schools] requires time, it requires effort, it requires availability. I come from the other side of Milan so probably tomorrow, I won't go back there once there is no foundation. To do this kind of work you need resources; if that piece is missing there [interviewer is referring to funding], it is difficult to maintain such a structure." Psychologist (sport club C)

As explained by the project manager, the challenge at the end of the three years of implementation was mainly related to the autonomy of football schools in working with the wider community. The challenge of transferring the skills and know-how of social workers to sports clubs, thus, emerges.

"The challenge is to make sure that this project is truly integrated within the territory and that it can then walk a little with its own legs. Because of the skills acquired by the soccer clubs; that still remains as a challenge, but I do not see it as a difficulty, that is, the fact that this territory is always constantly to be involved, the fact that there are potential issues and collaborations that we have not yet developed." Project Manager

One sport administrator pointed out the need of psychologists and educators in the sport club to work with coaches and parents in facing the challenges of youth during their sport path. Such professionals, however, are highly expensive and require specific funding that is not easy to find for a grassroots sport society.

"A psychologist attends our camps during training, during the games it is very important, (...) We had a psychologist, and they also need to be correctly paid, so I negotiated with them to reduce costs a little bit and then when I asked the sponsors to contribute. Many times, I didn't succeed [he means that he didn't manage to cover the costs of the psychologist]." Sport administrator (Sport Club C)

Thus, the necessity for psycho-social figures inside the sport clubs clashes with the theme of economic resources. The role of the foundation is, thus, fundamental for the sustainability of such figures. The challenge in this respect concerns the economic livelihood of this professional figure within the sports environment.

### 3.2.4. Lack of Sports Club Management Skills

Interviews highlighted the diversity and uniqueness of the participating sport clubs in terms of organizational culture and management. These unique qualities required implementation of ad hoc interventions adapted to each sports environment. In the initial implementation phase, the different organizational cultures and administrative approaches to sport management could have potentially facilitated/hindered engagement of the clubs in the program mission and activities.

Interviews in sport club A, for instance, showed how the lack of sport management skills could affect the impact of the program. At the time of the development of the program, sport club A had over 170 athletes. From a managerial point of view, sport club A had only two people (the president and secretary) delivering and handling the bureaucracy. Soccer training was delivered by an average of 30 volunteer coaches. The sport administrator was strongly committed and engaged with the social aims of the program because they were aligned with his personal values of inclusion that he practiced in the management of the club.

"When I talk about the boys of my teams, I get excited, I have been a volunteer for 30 years. Many people ask me "why do you do it?" but my "profit" is the fact that they give emotions that they don't even know they give. In every team we have one or two children from difficult contexts, not everyone has the opportunity to buy the sport equipment, sometimes teams have made collections to pay for the shirt or the jacket for those kids who cannot afford it." Sport administrator (sport club A)

The philosophy of "unconditional" inclusion, however, caused the sport administrator to host too many youths in sporting activities, even when they were not formally registered on the program. As a consequence, soccer training was attended by different young people every week sent by different stakeholders of the community. Most of the time, there was no communication of these new arrivals with the coach and social workers, who found themselves with new participants at soccer training.

In this regard, there was confusion and a lack of formality in the way the sport administrator managed the access of participants who were sent by different stakeholders in the community. However, this informal culture was not favorable for the daily work with the young people by the coach, the educator, and the psychologist. The openness toward the larger community and the great tension of this with the inclusion of marginalized people was, thus, not properly run, as reported by a program educator:

> "The soccer club has great capacity to welcome but not the ability to manage. Their policy is "we welcome all the people who come here, who are looking for a place to feel good, play and have fun."" Educator (sport club A)

The low quality of the management impacted the delivery of soccer training: there was a high turnover of sport coaches in program activities, and this affected athletes' feelings of mistrust toward the club. In the end, after one year, the program in this sport club was closed by the founders because of the lack of management skills.

On the contrary, in sport club B, the sensitivity towards pedagogical–educational issues by the sport administrators facilitated the fine tuning of the sports club to match the program's objectives:

> "Sport club B was a paradise, we couldn't believe it. They are well organized in reality but also very connected to pedagogical–educational logic. They are very sensitive and work within an area and with an approach which is already very close to the purposes of the program. Thus, we have not struggled to be in tune with them." Project Manager

> "Sport club B is a positive environment, I mean that they have a favorable approach [the interviewer is talking about the approach toward the program's scope]; inclusion is in their DNA." Psychologist (sport club B)

As the project manager explains, understanding and adapting to the uniqueness of each sporting reality was one of the biggest challenges of the project:

> "The difficulty has been to discover a diversity also linked to the characteristics of the people who are inside, that is, to discover that there are no common criteria that regulate sport and educational activities within the sport clubs, and this, at the beginning, surprised us a lot. I have to say that we had a slightly different representation, I thought that there were recognizable criteria with respect to management of the club, times, spaces, people, etc., instead, I found completely different realities and each made in its own way." Project Manager

At a managerial level, it emerged that the analysis of the characteristics of the sport clubs and of their local community created powerful conditions for the effective implementation of the program activities. However, such tailored work required a strong investment of time.

> "At the beginning of program implementation, we worked to gain knowledge and understanding of the sport clubs, people's knowledge, municipalities, of the context, of the main interlocutors. We spent four to five months thinking about these issues, but this is common for all community projects. You spend the first year doing this job, that is, building things and then you can work well in the following years" Project Manager

## 4. Discussion and Conclusions

The current research is original since it analyzes an Italian case for exploring the challenges of sport-based programs in promoting social inclusion [22]. Indeed, faced with many sports projects dedicated to promoting social inclusion, in Italy, there is still limited research dedicated to the topic [23]. This work highlights four challenges related to the usage of sport as a tool for social inclusion that suggest several insights at a theoretical level.

The first challenge is related to dropout of youth in living conditions of severe vulnerability; the high percentage of vulnerable youth in sport club C challenged the positive implementation of activities and gradually led to participants' dropout. This element of the research provides a warning about the impact of sport-based projects on highly vulnerable children. At a theoretical level, as reported also by Jeanes et al. [4], it is confirmed that sport can act on limited aspects of vulnerability but cannot fully impact the condition in a broad sense.

The second challenge is related to the limited transferability of program outcomes for youth in living conditions of severe vulnerability; our study found that, while young people were exposed to several positive benefits of sport, such as increased self-efficacy and acknowledgment of social norms, the intervention failed to effect behavioral changes at a wider social level, for instance at school, for youth living with marked social marginalization. This is in line with other studies of the field [27,32]. Drawing on Bailey's [5,6] theoretical contribution, the research reports that, when youth are living situations of extreme marginalization, outcomes on power dimensions are the most difficult to achieve [5,6]. Indeed, the research shows that the most vulnerable youth benefited little from taking more control over problematic aspects of their lives. At a theoretical level, data suggest that the impact of sport on social inclusion may vary according to the diverse grade of vulnerability of participants.

The third challenge is the limited sustainability of social workers; the research shows that the collaboration between sport and community actors helped to mitigate youth vulnerabilities by promoting insights and knowledge in the sport context and the wider community. This was helpful in order to strengthen the relationship among adults inside the network who maximized the emotional and social sustenance of the youngsters [25]. Monthly meetings with teachers actually impacted the comprehension of the adolescents in terms of their behaviors and attitudes in variant environments (school and sporting contexts). These features improved the holistic knowledge of the young people among the adults taking care of their educational path (teachers, social workers, sport workers).Although the work of interconnection between teachers and program educators and psychologists proved to be valuable for all the reasons mentioned, the sustainability challenge of social workers, who were the central nodes of this network, has been reported. The presence of social professionals indeed permitted bonds to form at the community level serving the social inclusion of youth. At a theoretical level, these results confirm that sport may affect the relational dimension of inclusion [5,6]. Young participants were indeed included in a network of collaborative adults who acted to promote their wider well-being. Such figures, however, are not taken for granted in amateur sports contexts, which are mainly based on voluntary work, at least in the Italian context [44]. The interviewees' experience suggests that sports clubs are struggling to find specific funds to pay these professionals. Furthermore, the presence of social workers partially obstructed the development of sports clubs' capacity to promote social inclusion on their own in collaboration with schools; this results in a paradox related to their presence in the sport context.

The last challenge is related to the lack of sports club management skills; the case study highlighted that some sport contexts may not be adequately equipped to manage social programs and to work in connection with other professions and sectors. Although working in connection with the community is an essential condition within sport-based programs promoting inclusion, sports clubs may not be prepared to manage this process. Nevertheless, the combination of sport clubs' conventions with social and educational aims can be challenging, if not adequately supported. In the cases under analysis, the

sport clubs' cultural openness toward wider social scopes facilitated implementation of the activities. This confirms the assumptions of several systemic and ecological theories sustaining how the micro-sport environment may strongly affect diverse development and social outcomes [32]. However, if sport clubs are not properly equipped in terms of management, as in the case of sport club A, the program risks failing in its aims. As pointed out by the program project manager, this work constitutes a challenge in terms of time and effort. The project manager's field experience highlights the need to work step by step with sports clubs to support them in implementing social inclusion processes. In this sense, the experience of sports club A is emblematic. The unstructured "inclusion" implemented by the sport management was, in fact, difficult to reconcile with the organizational culture of the foundation, which was much more focused on the codification and standardization of processes. This implies work of support, negotiation, and collaboration between the local actors and the foundation that is not always sustainable in terms of time and resources.

These challenges also suggest a number of implications for practice that should guide in designing and implementing sport-based programs for socially vulnerable youth.

The first one is related to the criteria for participants' recruitment [7]. Sport-based programs should equally involve youth living in diverse grades of vulnerability in order to avoid drop-out.

Second, with respect to the limited translation of sport outcomes in other life domains, a stronger involvement of the distal ecological systems [32] for supporting youth educational path outside sport should be considered. Several studies reported the benefit of a multi-stakeholder approach in sport, resulting in "economy of effort", namely, better use of resources and competencies for supporting youth [35,45–47]. The multi-stakeholder approach, thus, should be intensified, especially when working with vulnerable youth.

Third, the limited sustainability of social workers within sports clubs shows the paradox and complexity of working with vulnerable youth and remains unsolved. On the one hand, social workers are needed for promoting vulnerable youth social inclusion. Interviews indeed pointed out that educators and psychologists played the role of connectors with the community since they formed the relationship with the teachers outside the sports clubs. Schulenkorf [48] speaks of a "change agent" when describing such functions within the community. Change agents may be defined as an "anchor-person" acting as an external party who initiates contacts and facilitates cooperation among groups. According to this author, these figures generally have a crucial impact during planning and implementation phases of projects, as they support new contacts among groups and facilitate new collaboration.

The research confirms the cruciality of such figures in terms of social capital building at a local level serving the fragility of the youth. In more detail, the involvement of such professionals taking the role of "change agents" in opportunities for interorganizational work (e.g., monthly meetings, etc.) with community stakeholders (municipality representatives, teachers, community social workers, youth associations) promoted the discussion of youth vulnerabilities within and without the sporting environment and constituted a precious occasion to plan educational actions to support youth in need. Furthermore, concrete spaces of co-working, such as the monthly meetings, served as hubs for building "bridges" of collaboration among diverse actors [23]. On the other hand, the research also highlights that their presence obstructed the development of sport systems' capacity to include. Furthermore, sports clubs struggle in funding these figures.

Finally, the lack of management skills could be overcome through the implementation of specific training for sport coaches and administrators. Another possible strategy is the implementation of participatory approaches when planning and designing sport-based interventions on the behalf of donors [49,50]. This methodology, indeed, permits adaptation of actions and strategies in response to local contexts and needs. The implementation of participatory actions could be useful in order to draw the contextual sports clubs' characteristics, which eventually allows the creation of ad hoc paths promoting actions respecting local peculiarities [51,52]. Understanding local specificities at the planning stage,

indeed, could help intercept a lack of managerial skills on the part of sports clubs that results in a specific capacity, building pathways for those clubs that show themselves to be more fragile [49–56].

## 5. Limitations

First, the study is not generalizable.

Second, the research does not consider the point of view of participants, and this, of course, limits the understanding of the impact of the program at an individual level on youth and the challenges from their perspective. In this domain, future research should include young participants' engagement in the evaluation process.

Furthermore, the research cannot provide a conclusion about the long-term validity of the outcomes pointed out. In this domain, future research should encourage follow-up studies in order to understand what happens to the interorganizational network once the funding (or grant) has ended. Is the network of schools and sport clubs still functioning?

Future studies should also compare more cases in order to provide a wider understanding of the phenomena (8).

**Author Contributions:** C.D. and C.C. collected and analysed data; C.C. wrote the manuscript; C.D., C.C. and CG conceptually framed the manuscript; C.G. scientifically supervised the work. All authors have read and agreed to the published version of the manuscript.

**Funding:** This research received no external funding.

**Institutional Review Board Statement:** The study was conducted according to the guidelines of the Declaration of Helsinki. Ethical review and approval were waived for this study, due to the fact that author's institution didn't have an Etichs Committee at the time data were acquired.

**Informed Consent Statement:** Informed consent was obtained from all subjects involved in the study.

**Data Availability Statement:** Data supporting reported results are not available.

**Conflicts of Interest:** The authors declare no conflict of interest.

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
