# Peer review of "The Challenges of Promoting Social Inclusion through Sport: The Experience of a Sport-Based Initiative in Italy"

_societies, doi:10.3390/soc11020044_

Round 1

Reviewer 1 Report

  1. While your research and findings are interesting, the research is specific to this context making it more of a case study than a primary research paper.
  2. However, I think the topic of social inclusion, and promoting social inclusion through the initiatives of sport is very interesting.  If you can develop a more generalizable theory for this research topic and clearly articulate the theoretical contribution and managerial application, it will contribute academically and practically to this journal. 

Reviewer 2 Report

The work deals with a very important issue, rather difficult to implement as well as to interpret its results. And thus, its complexity appears to  cause problems at interpreting the findings properly in the Results part of the work.

From my point of view, it would increase the reader's interest if the article described in more detail the selected sports programs that were implemented to support the social inclusion of socially vulnerable youth and thus provided guidance for creating similar programs.

The study as it is presented suggests a collection of subjective opinions of people who participated in managing the programs, but their answers in such form cannot be generalized. I advise to process the results in a more transparent way, dividing them into the 3 basic areas presented in chapter 2.3 Data analysis and tools, or according to the role of individual participants (sport coaches, sport administrators, teachers; etc.). Without that, the results melt into chaotic interpretations.

Although the strengths and weaknesses of the project are mentioned in chapter 2.2 Procedures, they are missing in Discussion.

I further recommend that the criteria for including young people in the program be highlighted in a table; The inclusion criteria explained in the text sound and look rather informal and confusing.

The quality/comprehencibility of the article also suffers for denominating the participants using ambiguous abbreviations - only letters, as in line 524: "The main interlocutors for the program were V. and C". It is certainly possible to formulate a sentence in a way that abbreviations or even the full name of participant do not have to be used. This applies to the whole article.

The Abstract does not fully correspond with the essence of the problem discussed further; I propose to modify it accordingly.

Comments on formal insufficiencies:

Chapter 2 'Sport for vulnerable youth: an overview' is part of the theoretical analysis in Introduction. I suggest to avoid its numbering since the next chapter Materials and Methods is numbered 2, the same again, it is confusing.

 I suggest explaining, resp. correcting sources no. 23, 49 and 50 in References.

Reviewer 3 Report

Thank you very much for inviting me to review this wonderful article entitled: The challenges of promoting social inclusion through sport: The experience of a sport-based initiative in Italy.

First of all, I would like to congratulate the author(s) on this work. Social inclusion is very important in group sports, especially for young players. The author(s) provide some critical findings in the field. The reviewer can see the authors' hard works in the manuscript.

In general, I would like to say the contents of the study were reasonably presented. There has no big issue for the reviewer. Congratulations.

However, some minor suggestions may be provided to the author(s) for further consideration:

  1. The space of each paragraph should be using the same style.
  2. The first and second sections (Introduction and the overview) may be simplified to be easier for the readers. 
  3. It seems the author(s) didn't mention how many texts/manuscripts were produced/received from the interviews.
  4. Although the author(s) have mentioned their approaches on data trustworthiness and validity by using data triangulation, the results from the coding are highly recommended to provide to readers. The author(s) may provide a table or figure for your coding results, or provide it as an appendix.
  5. If there's any difficulty in providing the coding results, the frequencies of the themes may be depicted.
  6. Also, can the author(s) name the specific themes (or sub-categories) for their findings? Although the author(s) have provided three main findings as their themes based on the different perspectives, it seems still roughly performed.

Round 2

Reviewer 1 Report

  1. The discussion part is weak and the authors can provide more details about the relevance of the study and its link to similar studies. How is this study different than others?
  2. The study lacks cohesion in presenting ideas and significant findings, which also should clearly emphasize which findings or method usage of the research was new and original. Also the theoretical contribution to offer deeper insight into the literature should be clarified.

Author Response

Dear reviewer,

as you suggested, we provided more details about the relevance of the study and its link to similar studies. We further underlined the originality of certain findings and their theoretical contribution (line 577, 582,590, 610, 627)

Reviewer 2 Report

During the final editing, I suggest going through the grammar and stylistics of the text again (typos in the text).

Author Response

Dear reviewer,

as you suggested we revised the grammar of the manuscript.

Reviewer 3 Report

Great works! Congratulations.

Author Response

Thank you for the useful suggestions during the review process!